# Bovine Polyomavirus 2 is a Probable Cause of Non-Suppurative Encephalitis in Cattle

**DOI:** 10.3390/pathogens9080620

**Published:** 2020-07-29

**Authors:** Melanie M. Hierweger, Michel C. Koch, Torsten Seuberlich

**Affiliations:** 1Department of Clinical Research and Veterinary Public Health, Vetsuisse Faculty, University of Bern, 3010 Bern, Switzerland; melanie.hierweger@vetsuisse.unibe.ch (M.M.H.); michelkoch@gmx.ch (M.C.K.); 2School for Cellular and Biomedical Sciences, University of Bern, 3010 Bern, Switzerland

**Keywords:** bovine polyomavirus 2, non-suppurative encephalitis, neurological disease, cattle, next-generation sequencing, virus discovery, genome analysis, in situ hybridization

## Abstract

Tissues from two cows with neurological signs that were admitted to the Vetsuisse Faculty under suspicion of rabies and bovine spongiform encephalopathy (BSE), respectively, were further analyzed for this case report. After histopathological examination and exclusion of BSE and rabies, the animals were diagnosed with etiologically unresolved disseminated non-suppurative encephalitis. Using next-generation sequencing, we detected the full genome of bovine polyomavirus 2 (BoPyV2) in brain samples from both animals. This virus has been identified in beef samples in three independent studies conducted in the United States and Germany, but has not been linked to any disease. Analysis of the two new BoPyV2 genome sequences revealed close phylogenetic relationships to one another and to BoPyV2 isolates detected in beef samples. In situ hybridization demonstrated the presence of viral nucleic acid in all investigated brain areas and in areas with signs of inflammation in both animals. Thus, we provide the first evidence that BoPyV2 is a probable cause of non-suppurative encephalitis in cattle, and encourage further molecular and serological testing to elucidate the disease’s epidemiology, as well as experimental transmission studies to prove causality between the infection and disease.

## 1. Introduction

Polyomavirus infections are often persistent and subclinical, but their potential to cause severe diseases in immunosuppressed humans is well known. JC polyomavirus (JCPyV) and BK polyomavirus (BKPyV) are striking examples of viruses that become pathogenic in patients with AIDS and those under immunosuppressive treatment in the context of organ transplantation. Corresponding diseases, such as progressive multifocal leukoencephalopathy (PML), nephropathies, and hemorrhagic cystitis, often lead to fatal outcomes or transplant rejection, and thus play important roles in human medicine [1,2,3]. Polyomaviruses are also involved in the induction of tumors. The Merkel cell polyomavirus (MCPyV), for instance, can induce high-grade skin carcinomas in immunosuppressed people by integrating its genome into the host genome [4].

In immunocompetent individuals, the immune system controls most increases in viral activity. In the presence of immunosuppression, the different outcomes of polyomavirus infection are host- and tissue-dependent. The small (~5000 base pairs (bp)), circular, double-stranded DNA genomes of polyomaviruses are divided into three parts: an early region with open reading frames (ORFs) encoding early-expressed regulatory proteins (classically large T, small T, middle T/alternative large T (ALTO) antigens); a late region with ORFs encoding late-expressed structural proteins (classically VP1, VP2, and VP3); and a regulatory region containing promoters, transcription enhancers, and the origin for DNA replication [5]. The efficient production of T antigens—which stimulate the cell cycle and cell growth, and inhibit apoptosis—without productive viral replication or the transcription of late ORFs results in tumorigenesis. On the other hand, productive viral replication with sufficient production of structural proteins is linked to cell death and inflammation [6,7,8,9].

The International Committee on Taxonomy of Viruses classifies polyomaviruses into four genera: alpha-, beta-, gamma-, and delta-polyomaviruses. However, several polyomaviruses remain unassigned to a genus [10]. Polyomaviruses can also be classified according to genome features into VP3^+^ and VP3^–^ strains (with and without hints of VP3 expression, respectively) [11]. This division is relatively new, and focuses on the viruses’ phylogenetic relationships and putative common ancestors. MCPyV, a human polyomavirus with oncogenic potential, clusters with the VP3^–^ clade, which contains mainly animal polyomaviruses, including a recently discovered raccoon polyomavirus with tumorigenic potential and the newly discovered bovine polyomavirus 2 (BoPyV2) [11,12,13]. Bovine polyomavirus 1 (BoPyV1) is not in the VP3^–^ clade [11,13].

BoPyV1 was first detected in 1976 as a contaminant of fetal bovine serum used in cell culture systems [14,15,16,17]. The virus is very prevalent in the cattle population, and thus is often used as a parameter of environmental contamination by bovine excretion [18,19]. BoPyV1 seems to be shed from adult cattle, but isolation of the virus and ready viremia detection have been possible only in calves and bovine fetuses [20]. Serological studies revealed seroconversion in humans who have close contact with cattle [21]. However, no active human infection or association of this virus with any human or bovine disease has been documented.

BoPyV2 was first identified in beef samples in three unrelated studies. Its full length was determined in two viral metagenomics studies conducted around the same time in the United States [13,22], and by traditional sequencing in a study conducted in Germany [23]. We reported the identification of BoPyV2 in a brain sample from a cow with encephalitis of unknown origin; we sequenced the full virus genome using next-generation sequencing (NGS) and confirmed the result with conventional polymerase chain reaction (PCR) [24]. In that study, four additional neurologically diseased animals showed positivity on conventional PCR, but these results were inconclusive and not reproducible (unpublished observation and [24,25]), and no conclusion regarding a causal relationship of BoPyV2 infection with non-suppurative encephalitis could be drawn.

Here, we describe the full-length genome of BoPyV2, detected by NGS, in another animal with clear neurological signs and non-suppurative encephalitis, and compare the findings from this case with those from the first BoPyV2-positive case confirmed in our laboratory. Both animals had similar severe signs of non-suppurative encephalitis in the brain, and the sequenced BoPyV2 genomes are very similar. Additionally, we demonstrate the presence of virus in all brain areas examined by in situ hybridization (ISH), which indicates that BoPyV2 is a probable cause of neurological disease in cattle.

## 2. Results

### 2.1. Cases

#### 2.1.1. Case 34510

An approximately two-year-old dairy cow (animal ID 34510) from the canton of Bern, Switzerland, showed clinical signs suspicious of bovine spongiform encephalopathy (BSE), and was thus killed on 13 July 2002. No further information on clinical signs in this animal was available. BSE was excluded with the BioRad Platelia enzyme-linked immunosorbent assay (Marnes-la-Coquette, France) at the Division of Experimental Clinical Research, University of Bern, Switzerland. Histological examination revealed diffuse gliosis with glial nodules throughout the brain, but especially in the molecular layer of the cerebellum, as well as low-grade loss of Purkinje cells and low-grade perivascular infiltration of lymphocytes and a few macrophages in the brain parenchyma and meninges (Figure 1A–D). The animal was diagnosed with “non-suppurative encephalitis of unknown origin,” which encompasses a group of conditions with similar histological presentations and unknown, but presumably viral, etiologies.

#### 2.1.2. Case 51177

In the canton of St. Gallen, Switzerland, on 14 October 2017, an approximately two-year-old dairy cow (animal ID 51177) presented with neurological signs consisting of aggression, salivation, dysphagia, and separation from the herd. The animal was killed due to the suspicion of rabies. Rabies virus infection was excluded by immunofluorescence staining at the Swiss Rabies Center, Institute of Virology and Immunology, University of Bern, Switzerland. Additionally, the animal was tested for BSE with the Prionics Check Priostrip (Prionics AG, Schlieren, Switzerland) at the Division of Experimental Clinical Research, University of Bern, Switzerland; the result was negative. Histological examination revealed numerous subarachnoid, perivascular, and parenchymal bleeds and diffuse and multifocal nodular gliosis, especially in the gray, but also in the white, matter. Nodular gliosis was found around many degenerated or necrotic neurons. Mild perivascular cuffs with few lymphocytes, monocytes, and macrophages, sparse in the vascular wall, and sparse vessel-wall necrosis were also observed (Figure 1E,F). The diagnosis of “viral encephalitis” was made based on the morphological findings of severe, disseminated, lymphohistiocytic encephalitis with vasculitis, neuronal degeneration, and necrosis. The etiology of this case remains unresolved, but is presumed to be viral.

### 2.2. Determination of the Full BoPyV2 Genome

Results of NGS of a sample from animal 34510 have been described previously [24]. Briefly, paired-end (2 × 100 bp) sequencing of extracted DNA and RNA generated 82,396,767 and 57,746,275 reads, respectively. Mapping of the raw reads to the RefSeq viral genome database on 6 May 2015 resulted in coverage of the whole genome of the BPyV2-SF strain (accession no. KM111535.1), with mean read depths of 24.09 (DNA) and 26.32 (RNA). These results enabled the generation of the full circular genome of BoPyV2 (named BoPyV2 CH14, accession no. MT648389; Figure 2B. Because the NGS data suggested a 45-bp insertion in the ORF of the large T antigen compared with closely related BoPyV2 sequences, this inconclusive region was Sanger sequenced and the insertion was revealed to be an alignment mistake (binding sites of sequencing primers are shown in Figure 2A, orange squares). The length of the new genome was 5085 bp, with pairwise identities of 99.63% with BoPyV2b strain 2bS5 (accession no. KM496325.1) and 99.19% with BPyV2-SF.

NGS of RNA extracted from animal 51177 in paired-end mode (2 × 150 bp) generated 221,950,534 reads aligned to three scaffolds, with the greatest similarity to BoPyV2b isolate 2bS5. The scaffolds had lengths of 3637, 1097, and 836 bp and mean read depths of 58.74, 33.12, and 65.07, respectively. They covered the whole genome of BoPyV2, enabling manual assembly of a single circular 5085-bp sequence designated BoPyV2 CH17 (accession no. MT648388; Figure 2B). Analysis of the genome with BLASTn revealed that it was related most closely to BoPyV2b isolate 2bS5 andBoPyV2-SF, with pairwise identities of 99.63% and 99.27%, respectively.

NGS generated no plausible hit for a virus other than BoPyV2 for either animal. Hits for bacterial, fungal, and protozoal sequences are not reported in our bioinformatics pipeline because the brain pathology in both animals clearly suggested viral etiology.

BoPyV2 CH14 and BoPyV2 CH17 have 99.61% pairwise identity to each other. In both new BoPyV2 sequences, the typical ORFs of the polyomavirus family—those for early-expressed regulatory small-T, large-T, and ALTO proteins and late-expressed transcribed structural proteins VP1, VP2, and VP3—are present (Figure 2A). The VP3 ORF has no MALXXφ motif (φ represents an aromatic amino-acid residue) at its amino (N-) terminus, and the VP2 protein is surrounded by the strong Kozak sequence “TTT AGG ATG G” [26], in sequences from both animals. The regulatory and structural ORFs show contrary directions on the double-stranded DNA genome, separated by a regulatory region including six repetitions of GAGGC, which is an important motif for the DNA binding of T antigens [27].

### 2.3. Molecular Confirmation of BoPyV2

To confirm the presence of BoPyV2, PCRs to detect this virus were performed on samples from both animals. For animal 34510, the analysis and findings have been described previously [24]. For animal 51177, PCR of extracted DNA with the same primer pair used for BoPyV2 confirmation in animal 34510 (BPyV3/BPyV4) and an additional previously published primer pair (BPyV1/BPyV2) [13] yielded positive results (primer binding sites are shown in Figure 2A, orange squares). Results for non-template control samples were negative.

### 2.4. In Situ Detection of BoPyV2

ISH was performed with formalin-fixed paraffin-embedded (FFPE) brain tissue slides representing the cortex, midbrain, hippocampus, cerebellum, and brainstem for both animals, as well as the thalamus and caudate nuclei for animal 51177, for in situ detection of BoPyV2 viral nucleic acid (the binding region of the RNAscope probe is shown in Figure 2A, gray bar). Positive signals were expected to have predominantly nuclear or perinuclear localizations. Positive signals were detected in all available FFPE samples and were substantially stronger in samples from animal 34510 than in those from animal 51177 (Table 1). In comparison with signs of inflammation, BoPyV2 was detected most frequently in areas of gliosis, in glial nodules and perivascular cuffs (Figure 3), but also in areas with no apparent nodular gliosis or perivascular cuff. The positive staining coincided mainly with nuclei in samples from both animals (Figure 4). Negative control samples from animals with non-suppurative encephalitis of other origins did not show positive nuclear staining.

## 3. Discussion

In this report, we present the cases of two cows with non-suppurative encephalitis of unknown origin and neurological symptoms. The animals had similar histopathological features of disseminated, nodular gliosis involving the entire brain, neuronal loss, and sparse perivascular cuffs consisting mainly of lymphocytes. No parallel could be drawn from the clinical presentations of the animals due to the sparse clinical information for animal 34510. Nevertheless, full BoPyV2 genomes were obtained by NGS of brain samples from both cows; they differed only slightly from one another and from previously described BoPyV2 sequences. Thus, the new genomes represent new strains of BoPyV2, designated BoPyV2 CH14 (sequenced from animal 34510) and BoPyV2 CH17 (sequenced from animal 51177). The general organization of the new BoPyV2 sequences is in accordance with that of published BoPyV2 sequences [13,22], and the sequences show typical features of the polyomavirus VP3^–^ clade, such as a strong Kozak sequence around the start codon of VP2 and no MALXXφ motif at the N-terminus of VP3 [11,13]. These findings are not surprising, as the new BoPyV2 genomes are highly (>99%) similar to previously described and analyzed BoPyV2 sequences. Polyomaviruses are thought to co-evolve with their hosts, which is in accordance with the close phylogenetic relationships of hosts of most members of the VP3^–^ clade [11]. However, transmission from one host of the VP3^–^ clade is more plausible for BoPyV2 because cattle are phylogenetically distant from other VP3^–^ hosts, making co-evolution unlikely. Additionally, polyomaviruses from one host species are usually very similar to each other. BoPyV1 does not belong to the VP3^–^ clade and is only distantly related to BoPyV2, suggesting recent inter-species transmission of BoPyV2. Although these observations point to the potential for BoPyV2 transmission to phylogenetic distant hosts, this theory must be considered with caution, as the assumptions are based purely on phylogenetic observations [11,13,22].

We detected BoPyV2 viral nucleic acids in situ in the brains of both cows that we examined. With the RNAscope probe, viral RNA transcripts and DNA can be detected (RNAscope technical support, personal communication). As expected, most viral staining colocalized with nuclei, where polyomaviruses replicate in the cell. Some positive staining was also seen in the vicinity of nuclei because viral proteins are translated from mRNA transcripts at ribosomes in the cytoplasm [5]. Positive cells were often observed in close association with pathological lesions that are typical for a non-suppurative encephalitis [28]. This finding is important because BoPyV2 has been found in beef samples [13,22,23] and bovine encephalitis cases [24], but has not been linked clearly to any disease in cattle. According to Lipkin’s [29] proposed hierarchy of confidence for the evaluation of pathogen disease causality, the detection of viral nucleic acid in diseased tissue indicates a possible correlation of the virus with the corresponding disease, and an in situ finding at sites with signs of inflammation indicates a probable correlation. In addition, the detection of one pathogen in many diseased animals strengthens the assessment. As we have detected BoPyV2 in only a few cases of bovine encephalitis to date, we cannot yet assess the statistical significance of the association of this virus with disease. Interestingly, BoPyV2 was not found in 50 control animals with no sign of encephalitis in the brain [24], and was the only virus detected by NGS in the animals described in this case report.

Our ability to sequence the complete BoPyV2 genomes from RNA suggests active replication of the virus in the animals’ brains. NGS read coverage was similar for early and late viral ORFs, suggesting that BoPyV2 causes lytic infection, in accordance with the brain histopathological findings for both animals. Given the in situ detection of BoPyV2 throughout the brains of the animals that we examined and previous findings for this virus from beef samples, the infection route may implicate viremia, with infection of the brain by viruses crossing the blood–brain barrier. As the blood–brain barrier is generally leaky in encephalitis [30], secondary entry of the virus into the already-diseased brain cannot be excluded at this point. We cannot exclude an underlying causative infection with an undetected highly divergent virus in our cases, despite the negative NGS findings. Furthermore, the disease stage was unknown in our cases; the causative pathogen(s) might have been cleared by the immune system in the brains at the time of sampling. The histopathology in both cases strongly suggests a viral etiology, but an underlying infection with other pathogens (i.e., bacteria, protozoa, or fungi) cannot be excluded and would have been missed by our bioinformatics pipeline. Hence, the incidental nature of the finding of BoPyV2 in the diseased brains cannot ultimately be ruled out. Latent infections in lymphoid tissue and reservoirs in inflammatory monocytes have been described for other polyomaviruses [31,32,33]; latently infected and released peripheral blood leukocytes could be disease-unrelated sources of positivity in the brain parenchyma, although latent infection seems unlikely because of the detection of abundant viral RNA.

Other polyomaviruses have been shown to cause neurological diseases. Recently, a novel raccoon polyomavirus related closely to BoPyV2 was identified as the cause of brain tumors in wild-ranging raccoons in the United States [12]. JCPyV is known to cause PML, a lytic infection of the central nervous system (CNS) white matter, in immunosuppressed humans [1,34]. The histological presentation of this disease is dominated by demyelination due to oligodendrocyte lysis, which differs from that seen in the two cows described in this report. However, the JCPyV has been found to cause a spectrum of CNS diseases, including granule cell neuropathy and encephalopathy (infections of the cerebellar and cerebral gray matter, respectively) and meningitis, in immunosuppressed humans [1,35,36]. Gray-matter infections with resulting neuronal loss are typically characterized by surrounding reactive gliosis accompanied by sparse perivascular infiltrates. These histopathological findings are more similar to the lesions observed in the two cows than are classical PML lesions. In addition, BKPyV rarely induces meningoencephalitis, affecting mainly ventricular and meningeal surfaces, in immunosuppressed patients [37]. The SV40 virus can also induce meningoencephalitis; on experimental infection, monkeys that were positive for simian-human immunodeficiency virus developed a PML-like phenotype [38] or productive neuronal infection [39]. MCPyV, which is related closely to BoPyV2, has been associated with gliomas [40]. These examples show the potential of polyomaviruses to cross the blood–brain barrier and/or cause neurological disorders. Whether immunosuppression is also relevant for active BoPyV2 infection, and whether the animals investigated in this study were immunosuppressed, unfortunately are not known. Metabolic imbalances occurring during the transition from pregnancy to lactation can have immunosuppressive effects, especially in dairy cows [41], and plausibly predispose cows to BoPyV2 infection. Immunosuppression on infection with other pathogens, such as bovine virus diarrhea virus [42], also cannot be excluded, as non-nervous tissues and blood samples were not available for investigation in our cases.

Overall, we conclude that BoPyV2 infection is a probable cause of non-suppurative encephalitis in cattle, but we emphasize the need for further research on this virus. Targeted testing of many more encephalitis cases is needed to estimate the infection prevalence and statistical association of BoPyV2 with this disease in cattle. The investigation of other bovine tissues and blood samples is needed to identify putative virus reservoirs in the body, and to assess the infection route and prevalence of the virus in the cattle population. In vitro studies will be crucial to determine the virus’s pathological potential in different neuronal cell types and for further interpretation of our results. Definite proof of causality between BoPyV2 and non-suppurative encephalitis in cattle, however, requires experimental animal infection studies. Nevertheless, based on our findings, we encourage the inclusion of BoPyV2 infection in the differential diagnosis and in testing schemes for unresolved bovine encephalitis cases.

## 4. Materials and Methods 

### 4.1. Samples

Fresh-frozen brain tissue from animal 51177 was available for nucleic acid extraction and subsequent molecular testing. The material included one brain hemisphere. FFPE brain tissue slides from the other hemisphere, including the cerebral cortex, midbrain, thalamus, hippocampus, cerebellum, brainstem, and caudate nuclei, were available for in situ investigations. For animal 34510, in situ investigations were performed on FFPE material from the cerebral cortex, midbrain, hippocampus, cerebellum, and obex. FFPE brain tissue slides from BoPyV2-negative animals with non-suppurative encephalitis (animal IDs 33181 and 44282) [24] served as negative controls for all in situ investigations.

### 4.2. Nucleic Acid Extraction

For DNA and RNA extraction for animal 51177, pooled frozen tissue samples of the frontal cortex, thalamus, and brainstem were used. DNA was extracted with the DNeasy Blood and Tissue Kit (Qiagen, Hilden, Germany), and RNA was extracted with TRI Reagent (Sigma Life Sciences, St. Louis, Missouri, MO, USA), according to the manufacturers’ instructions.

### 4.3. NGS

For NGS, RNA from animal 51177 was used. cDNA was produced using the SMARTer Ultra Low Input RNA Kit (Takara Bio Inc., Kusatsu, Japan), and library preparation was performed with the TruSeq Stranded Total RNA Kit (Illumina, San Diego, CA, USA), as described by the manufacturers. The library was then sequenced in paired-end mode (2 × 150 bp) on an Illumina HiSeq 3000 machine (Illumina, San Diego, CA, USA). NGS data are available in the National Center for Biotechnology Information sequence read archive (BioProject ID: PRJNA639014).

### 4.4. Bioinformatics Analysis

After quality control with fastqc (version 0.11.7), quality trimming with trimmomatic (version 0.36), and mapping to the bovine reference genome (BioProject ID PRJNA32899, Bos_taurus_UMD_3.1) with STAR (version 2.6.0c), unmapped reads were assembled to scaffolds using SPAdes (version 3.12.0; settings: --meta --k 21,33,55,77). Alignment of the scaffolds to viral databases of GenBank and RefSeq viral nucleotide sequences and UniProt viral amino-acid sequences downloaded on 25 July 2018 was then performed using BLASTn (version 2.7.1+) and DIAMOND (version 0.9.18) with default settings. Scaffolds with hits for >10% of their lengths at the nucleotide level with sequences in an in-house non-viral database (including archaeal, bacterial, fungal, mammalian, and protozoan sequences), and those with longer hits with sequences in this non-viral database than with those in the viral databases, were excluded from further investigation. Complementary sequencing data analysis was performed using Geneious software (version 9.1.8; Biomatters, Auckland, New Zealand).

### 4.5. PCR

PCR of DNA extracted from animal 51177 was performed in 1× GoTaq^®^ Green Master Mix (Promega Corporation, Madison, WI, USA) with 400 nM of each previously published primer (BPyV1 and BPyV2 or BPyV3 and BPyV4 [13]) and 2 μL DNA in an end volume of 25 μL. Cycling conditions were set to 94 °C for 2 min, followed by 35 repetitions of denaturing at 94 °C for 30 s, annealing at 50 °C (for BPyV1/BPyV2) or 63 °C (for BPyV3/BPyV4) for 30 s, and elongation at 72 °C for 30 s, and a final elongation step at 72 °C for 7 min. In both PCRs, water served as a negative control. The PCR products were analyzed by gel electrophoresis on a 1% agarose gel.

### 4.6. Sanger Sequencing

For Sanger sequencing of samples from animal 34510, primers spanning a 400-bp-long fragment of the ORF of the large T antigen were designed based on NGS data using Geneious software (version 9.1.8). Sanger sequencing was performed with previously extracted RNA and DNA [24]. First, the RNA was reverse transcribed to cDNA using SuperScript™ III reverse transcriptase (Thermo Fisher Scientific Inc., Waltham, MA, USA) and random hexamers, as described by the manufacturer. Then, cDNA and DNA were amplified with Q5^®^ Hot Start High-Fidelity DNA Polymerase (New England Biolabs Inc., Ipswich, MA, USA) using a standard protocol, forward primer BoPyV2-CH14_826F (TTCAGATGTGGGACCGGCT), reverse primer BoPyV2-CH14_1125R (TGTCTTTGGTGGAGTACTGTCA), and an annealing temperature of 66 °C. The PCR products were run on a 1% agarose gel, cut out, and purified with the NucleoSpin^®^ Gel and PCR Clean-up Kit (Macherey-Nagel, Düren, Germany) according to the manufacturer’s protocols. Subsequent Sanger sequencing was performed on a 3730 DNA Analyzer (Thermo Fisher Scientific Inc., Waltham, MA, USA) using the BigDye^®^ Terminator v3.1 Cycle Sequencing Kit (Thermo Fisher Scientific Inc., Waltham, MA, USA), about 100 ng purified PCR product, the forward and reverse primers, and the protocol described by the manufacturer. The data obtained were analyzed using Geneious software (version 9.1.8; Biomatters, Auckland, New Zealand)

### 4.7. ISH

For ISH with FFPE tissue slides for animals 34510 and 51177, an RNAscope probe targeting the positive strand of the BoPyV2 CH17 major capsid protein (VP1) ORF (nucleotide positions 2–1152; cat no. 571941; Advanced Cell Diagnostics, Newark, NJ, USA) was designed. This probe has only four mismatches with the positive strand of the VP1 ORF of BoPyV CH15 (at nucleotide positions 2834, 2969, 3494, and 3611 of the genome), and thus is thought to bind both viral genomes. Chromogenic staining of all available native FFPE tissue slides from animals 34510 and 51177 was performed using the RNAscope 2.5 Detection Kit-Brown (Advanced Cell Diagnostics) according to the manufacturer’s instructions. For counterstaining, Mayer’s hemalum solution (Merck KGaA, Darmstadt, Germany) was used; the slides were then mounted with Aquatex^®^ media (Merck KGaA, Darmstadt, Germany) and analyzed under a Zeiss Axio Scope.A1 (Carl Zeiss Microscopy GmbH, Göttingen, Germany).

For each FFPE tissue slide, the severity of non-suppurative inflammation and quantity of ISH-positive cells were assessed semi-quantitatively. Signs of inflammation were graded as absent, mild (only scattered inflammatory cells observed), moderate (multifocal), and severe (disseminated). ISH staining was graded as negative, weakly positive (scattered positive cells in the brain parenchyma), moderately positive (multiple positive cells), and strongly positive (abundant positive cells).

Additionally, fluorescent in situ hybridization was performed with FFPE tissue slides from the cerebral cortex of animal 51177 and the cerebral cortex and hippocampus of animal 34510, using the RNAscope 2.5 Detection Kit-Red (Advanced Cell Diagnostics) and the probe for BoPyV2 (Advanced Cell Diagnostics) as described by the manufacturer. The slides were counterstained with DAPI BioChemica (AppliChem GmbH, Darmstadt, Germany) diluted 1:10,000 in phosphate-buffered saline with Tween for 1 h at room temperature in humid conditions, and then mounted with Glycergel^®^ Aqueous Mounting Medium (Dako Denmark A/S, Glostrup, Denmark). The slides were analyzed under an Olympus Fluoview FV3000 confocal laser scanning microscope (Olympus Europa, Hamburg, Germany). For both ISH staining methods, native FFPE brain tissue slides from animals 33181 and 44282 served as negative controls.

## Figures and Tables

**Figure 1 pathogens-09-00620-f001:**
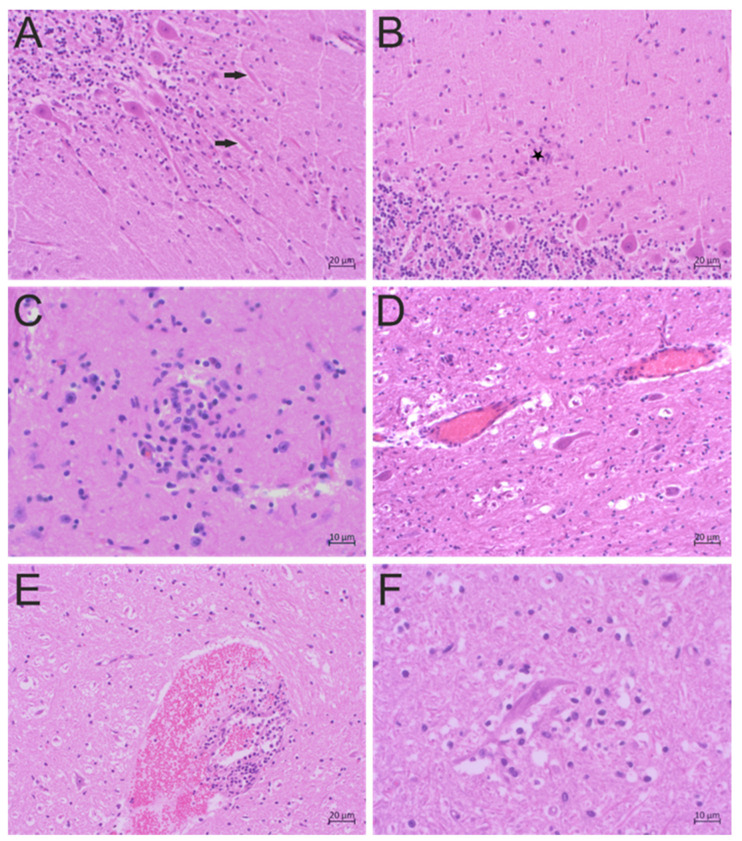
Histopathological findings of disseminated non-suppurative encephalitis in two neurologically diseased cows. (**A**–**D**) Hematoxylin and eosin (H&E) staining of brain sections from animal 34510. (**A**) Reactive gliosis around Purkinje cells and axonal swelling (arrows) in the cerebellum, (**B**) a glial nodule (asterisk) in the molecular layer of the cerebellum, (**C**) a glial nodule in the midbrain, (**D**) mild perivascular cuffs and diffuse gliosis in the cerebral cortex. (**E**,**F**) H&E staining of brain sections from animal 51177. (**E**) Perivascular bleeds with infiltration of the vessel wall and perivascular space, mainly by lymphocytes, in the cerebral cortex; (**F**) nodular gliosis in the cerebral cortex.

**Figure 2 pathogens-09-00620-f002:**
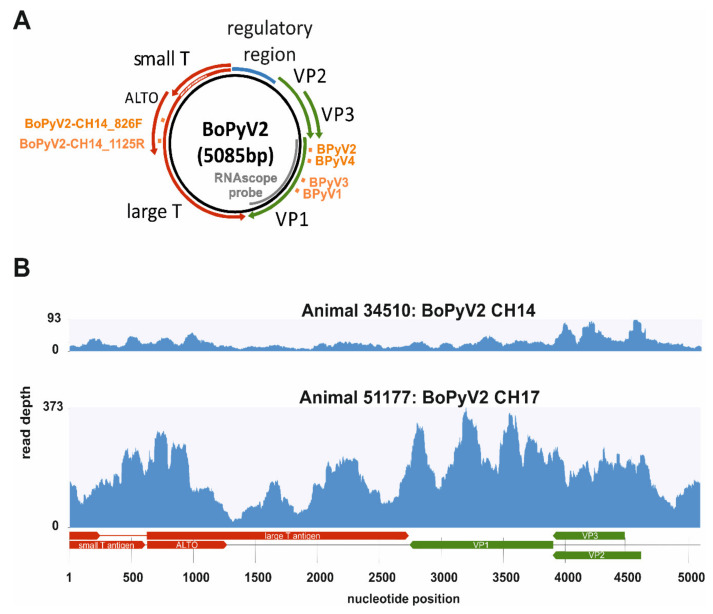
Schematic representation of the genomic structure of bovine polyomavirus 2 (BoPyV2) and the coverage of reads obtained from next-generation sequencing (NGS). (**A**) The double-stranded genome of BoPyV2 is depicted in a black circle, the regulatory region is illustrated by a blue box, and the open reading frames (ORFs) are illustrated by arrows (red, ORFs of early-expressed proteins; green, ORFs of late-expressed proteins). Primer binding sites are depicted by orange squares and the BoPyV2 RNAscope probe is represented by a gray bar. (**B**) The whole genome of BoPyV2 is covered by reads generated from NGS of samples from animals 34510 (BoPyV2 CH14) and 51177 (BoPyV2 CH17). The read depth at each nucleotide position of the genome is given.

**Figure 3 pathogens-09-00620-f003:**
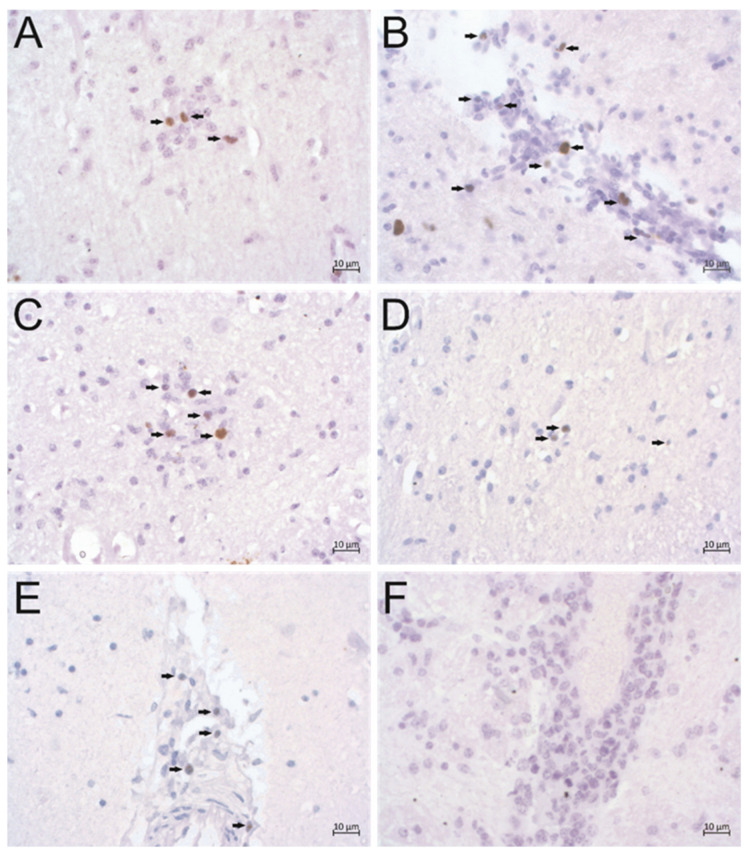
In situ detection of bovine polyomavirus 2 (BoPyV2) nucleic acid. Chromogenic in situ hybridization (ISH) was performed with formalin-fixed paraffin-embedded (FFPE) brain tissue slides using the BoPyV2 RNAscope probe. Positive signals (brown, arrows) can be seen in areas with signs of inflammation. (**A**) Glial nodule in the molecular layer in the cerebellum of animal 34510, (**B**) perivascular cuff in the midbrain of animal 34510, (**C**) glial nodule in the midbrain of animal 34510, (**D**) gliosis in the thalamus of animal 51177, (**E**) perivascular cuff in the thalamus of animal 51177. (**F**) Signs of inflammation on FPPE brainstem tissue slides from negative control animal 44282 (perivascular cuff) without positive ISH staining.

**Figure 4 pathogens-09-00620-f004:**
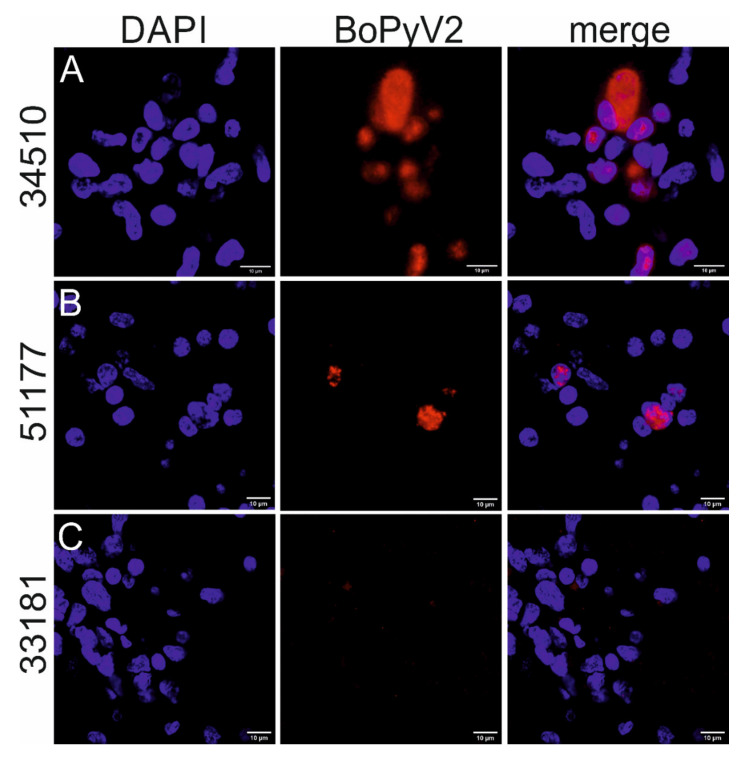
Associations of bovine polyomavirus 2 (BoPyV2) nucleic acids with cell nuclei. Fluorescent in situ hybridization was performed with formalin-fixed paraffin-embedded brain tissue slides using the BoPyV2 RNAscope probe. Positive signals (red) are colocalized with nuclei (blue). (**A**) Cerebral cortex of animal 34510, (**B**) cerebral cortex of animal 51177. (**C**) The thalamus of negative control animal 33181 shows no positive staining colocalized with nuclei. Scale bars at the bottom of each micrograph correspond to 10 µm.

**Table 1 pathogens-09-00620-t001:** Severity of histopathological lesions and viral RNA detection by in situ hybridization.

Animal	Brain Region	Histopathological Lesions	ISH
34510	Cerebral cortex	++	3
	Midbrain	+++	3
	Hippocampus	++	3
	Cerebellum	+++	3
	Obex	+	2
51177	Cerebral cortex	++	1
	Midbrain	++	2
	Hippocampus	+	1
	Cerebellum	++	2
	Brainstem	++	1
	Caudate nuclei	++	1
	Thalamus	+	2

ISH, in situ hybridization; + mild, ++ moderate, +++ severe, 1: weakly positive, 2: moderately positive, 3: strongly positive.

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
