# Peer review of "Bovine Polyomavirus 2 is a Probable Cause of Non-Suppurative Encephalitis in Cattle"

_pathogens, 2020, doi:10.3390/pathogens9080620_

Round 1

Reviewer 1 Report

General comments

This manuscript describes the discovery and analysis of bovine polyomavirus 2 in two cows suffering from neurologic disease.  Using genetics and pathology, the authors argue that BoPyV2 is a putative cause of non-suppurative encephalitis in cattle, and not simply an incidental finding in these cases.  The combination of metagenomics and histopathology, both of which are well done, is a strength of this case report.  The figures are also very good.  The manuscript is well organized but it would benefit from an additional round of English language editing.

My main critique is with the Discussion, which tries to argue too hard that BoPyV2 is the cause of this disease.  The authors should discuss other possibilities, such as an underlying condition that predisposes animals to infection with BoPyV2.  The authors should also state what other viruses and bacteria (if any) were present in their metagenomic sequence data

Specific comments

Line 15.  “Relatively new” is unclear.  Perhaps “recently discovered” or “recently described.”  However, it appears from reference 13 that BoPyV2 was first reported in 2014, which is not recent.  Consider deleting this phrase and other similar phrases throughout.

Line 20 “In the entire brain” is not clear.  The authors should be more precise here.

Lines 22-23.  Molecular and serologic testing are unlikely to resolve causality.  Experimental infection would be the most definitive study design.

Line 40: “Lie within the virus’ molecular biology” is not clear.

Line 63: It is not clear what is meant by virus isolation being detected.

Line 78, 99: “Symptoms” is reserved for humans.  Animals show “signs.”

Line 81: “Widespread in both brains” is too general.  I suggest being more specific without getting into too much detail (see comment below).

Figure 2A: the genome looks like an oval, but it should be circular.

Lines 123-143.  The authors should state what other microorganisms were detected in their sequence data.  It would be very unusual to detect only this one virus and no other viruses, bacteria, protozoa, etc.

Lines 173-183.  The authors should state clearly which parts of the brain were positive and which parts of the brain were negative.  It is unclear whether cortex, midbrain, hippocampus, thalamus, cerebellum and brainstem were all positive or whether only cortex, midbrain and thalamus were positive.

Lines 224-252.  This paragraph is very long and should be divided and reorganized to flow more smoothly.

Lines 275-283.  As mentioned above, only experimental infection studies will definitively prove or disprove causation.  The authors should discuss this, as well as alternative explanations for why the finding of BoPyV2 might still be incidental – for example, an underlying condition could cause both encephalitis and infection with BoPyV2.

Lines 346-348.  The authors should state whether the probe was an exact match to both BoPyV2 sequences and , if not, should provide the sequence of the probe.

Author Response

Reviewer 1:

Point: "Line 15.  “Relatively new” is unclear.  Perhaps “recently discovered” or “recently described.”  However, it appears from reference 13 that BoPyV2 was first reported in 2014, which is not recent.  Consider deleting this phrase and other similar phrases throughout."

Response: Thank you for your comment. We deleted this phrase.

Point: "Line 20 “In the entire brain” is not clear.  The authors should be more precise here."

Response: We exchanged the phrase to "all tested brain areas" and made these areas clear for both cases in the materials and methods and results section.

Point: "Lines 22-23.  Molecular and serologic testing are unlikely to resolve causality.  Experimental infection would be the most definitive study design."

Response: This is true! We made clear that the former tests do not prove causality and only experimental infection can do so.

Point: "Line 40: “Lie within the virus’ molecular biology” is not clear."

Response: Thank you for your comment. We deleted the phrase.

Point: "Line 63: It is not clear what is meant by virus isolation being detected."

Response: We rewrote the phrase to "… but it was only possible to isolate virus and readily detect viremia in calves and bovine fetuses"

Point: "Line 78, 99: “Symptoms” is reserved for humans.  Animals show “signs.”"

Response: We exchanged the word "symptoms" by the word "signs".

Point: "Line 81: “Widespread in both brains” is too general.  I suggest being more specific without getting into too much detail (see comment below)."

Response: We again exchanged the phrase to "all tested brain areas" to be more specific and provide details in materials and methods and results.

Point: "Figure 2A: the genome looks like an oval, but it should be circular."

Response: Thank you we adjusted the figure.

Point: "Lines 123-143.  The authors should state what other microorganisms were detected in their sequence data.  It would be very unusual to detect only this one virus and no other viruses, bacteria, protozoa, etc."

Response: No other viruses were detected in our sequencing data. Bacterial, protozoal, fungal, archeal and mammal sequences are excluded from further investigations with our settings because we expect a viral etiology in the cases of this study. We also explained this in the manuscript.

Point: "Lines 173-183.  The authors should state clearly which parts of the brain were positive and which parts of the brain were negative.  It is unclear whether cortex, midbrain, hippocampus, thalamus, cerebellum and brainstem were all positive or whether only cortex, midbrain and thalamus were positive."

Response: All tested brain areas were positive. We listed the positive brain areas in the result section and added a table with a semi-quantitative evaluation of the positive ISH results and the degree of inflammation.

Point: "Lines 224-252.  This paragraph is very long and should be divided and reorganized to flow more smoothly."

Response: We divided the paragraph.

Point: "Lines 275-283.  As mentioned above, only experimental infection studies will definitively prove or disprove causation.  The authors should discuss this, as well as alternative explanations for why the finding of BoPyV2 might still be incidental – for example, an underlying condition could cause both encephalitis and infection with BoPyV2."

Response: We made clear that only experimental infection could prove the causality. Furthermore, we added alternative explanations of unrelated and undetected infections of the brain and discussed reasons for an underlying immunosuppression, which could have led to a BoPyV2 infection.

Point: "Lines 346-348.  The authors should state whether the probe was an exact match to both BoPyV2 sequences and , if not, should provide the sequence of the probe."

Response: The probe is an exact match to BoPyV2 CH17 and shows 4 mismatches to BoPyV2 CH15. We made this clear in the materials and methods.

Reviewer 2 Report

The paper describes the detection of a new polyomavirus of cattle and its  association to non suppurative encephalitis. The study is well presented and describes deeply the presence of both genome and viral particles in the brain tissue of two cattle.

 Have you tested the sample 51177 for BoAstV, PIV-5 BoRV, BHV-6, OvHV-2 as you did for sample 34510 using PCR, RT-PCR or real time PCR?Why you have not considered Schmallenberg virus, Bluetongue virus and Bovine Viral Diarrhea virus for differential diagnosis?

Why did you start from RNA and not from DNA for 51177 sample? Could this affect the ability of  NGS to detect latent DNA virus present in the brain tissue?

Could you hypothesize a viral mechanism of action in determining the histopathological effects observed in brain tissue?

Author Response

Reviewer 2:

Point: "Have you tested the sample 51177 for BoAstV, PIV-5 BoRV, BHV-6, OvHV-2 as you did for sample 34510 using PCR, RT-PCR or real time PCR?Why you have not considered Schmallenberg virus, Bluetongue virus and Bovine Viral Diarrhea virus for differential diagnosis?"

Response: No, we have not tested the sample 51177 for the listed viruses using PCR, RT-PCR or real time PCR because we expect that we had found these viruses with NGS with the applied parameters. We did not consider Schmallenberg virus, Bluetongue virus and Bovine Viral Diarrhea virus for differential diagnosis because we did not get hints of an involvement of these viruses with NGS.

Point: "Why did you start from RNA and not from DNA for 51177 sample? Could this affect the ability of  NGS to detect latent DNA virus present in the brain tissue?"

Response: We started from RNA because in the study of Wüthrich et al., in which we sequenced RNA and DNA, we were able to detect all DNA viruses with RNA sequencing. It cannot be excluded for sure that latent DNA viruses are missed with solely RNA sequencing, but we expect relevant latent viruses to transcribe latency associated ORFs, which should be detected by RNA sequencing.

Point: "Could you hypothesize a viral mechanism of action in determining the histopathological effects observed in brain tissue?"

Response: Unfortunately, from the data we have obtained until now, we do not feel competent to hypothesize on a viral mechanism. More cases of bovine encephalitis with an in situ confirmed BoPyV2 involvement have to be investigated and/or in vitro infection studies have to be performed to investigate the viral mechanism.

Reviewer 3 Report

Hierweger and collaborators report a possible causal association between BoPyV2 and non suppurative encephalitis in two cattle in Switzerland.

To better clarify the cases and methodology used some minors will be needed:

Line 51 according to ICTV it would be better to mention also species unassigned to the genus besides alpha, beta, gamma  and delta-polyomaviriuses

Line 53 please include a reference in support of the VP3+ and VP3- classification

Cases description #3451 and 51177 since BoPyV2 has been previously reported in beef, it would be interesting to know the animals were dairy or beef cattle

Human Polyomaviruses such as JCPyV are known to cause CNS disease in immunosuppressed patients. At Line 272 the authors state that  an immunosuppression may be also relevant for an active infection of BoPyV2. A common cause of immunosuppression in cattle is BVDV, did the authors attempt to test the animals for BVDV?

Line 286-292 the authors describe the samples: Fresh frozen brain tissue was available for nucleic acid extraction for the animal 51177while FFPE tissue was used for # 34510 and negative controls. Did the Authors evaluate nucleic acid viability for these samples before testing? If yes how

Author Response

Reviewer 3:

Point: "Line 51 according to ICTV it would be better to mention also species unassigned to the genus besides alpha, beta, gamma  and delta-polyomaviriuses"

Response: Thank you for your comment. We mentioned the unassigned polyomavirus species in the manuscript.

Point: "Line 53 please include a reference in support of the VP3+ and VP3- classification"

Response: We added the reference to better clarify the citation.

Point. "Cases description #3451 and 51177 since BoPyV2 has been previously reported in beef, it would be interesting to know the animals were dairy or beef cattle"

Response: Both animals were dairy cattle. We added this to the case description.

Point: "Human Polyomaviruses such as JCPyV are known to cause CNS disease in immunosuppressed patients. At Line 272 the authors state that  an immunosuppression may be also relevant for an active infection of BoPyV2. A common cause of immunosuppression in cattle is BVDV, did the authors attempt to test the animals for BVDV?"

Response: We did not specifically test the animals for BVDV because we did not have any blood samples or additional tissue samples except of the brain. We made this clear and discussed an infection with BVDV as a possible reason for immunosuppression.

Point: "Line 286-292 the authors describe the samples: Fresh frozen brain tissue was available for nucleic acid extraction for the animal 51177while FFPE tissue was used for # 34510 and negative controls. Did the Authors evaluate nucleic acid viability for these samples before testing? If yes how"

Response: In this study, we only extracted DNA and RNA of animal 51177 to perform NGS and PCRs. The viability of the RNA was tested before library preparation for NGS according to standard procedures. The viability of DNA was not tested. FFPE material of animals 51177, 34510 and negative control animals was solely used for in situ investigations. We explained this better in the materials and methods section.